# OpenReview forum: "Eliciting Behaviors in Multi-Turn Conversations"
_ICLR.cc/2026/Conference — Submitted to ICLR 2026_

### Official Review · Reviewer_HJpX · 2025-10-31

**Soundness:** 2
**Presentation:** 2
**Contribution:** 2
**Rating:** 4
**Confidence:** 4

**Summary:**

This paper tackles automated behavior elicitation for multi-turn LLM conversations, arguing static benchmarks are insufficient. It introduces a framework categorizing methods into prior, offline, and online interaction, and proposes EMBER, a multi-turn online RL variant. Across three tasks and multiple target models, five methods are compared: online methods achieve the highest average success (~36%) with a few thousand interactions, offline approaches show cross-objective generalization, and EMBER uncovers failure modes missed by single-turn. Qualitative analyses indicate that online methods commonly discover systematic failure patterns that generalize beyond the training examples.

**Strengths:**

1. The paper clearly formulates multi-turn behavior elicitation by casting each method as learning a prompt distribution under a verifiable rubric, which standardizes cross-method comparison and improves methodological clarity.
2. The analytical framework that organizes approaches into prior, offline, and online families is timely and useful, making the design space explicit and guiding the choice of evaluation protocols.
3. EMBER extends online RL-style optimization to multi-turn conversations with practical design choices such as interleaved rollouts, repetition penalties, and strategy-to-message factorization, which make exploration in dialogue settings feasible.
4. The work addresses an important and timely problem—evaluating LLMs in realistic multi-turn dialogue—and motivates adaptive evaluation protocols that can impact red-teaming and safety practices.
5. The insights into how prior knowledge, training-set diversity, and number of turns influence performance are original and practically valuable for designing future evaluation pipelines.

**Weaknesses:**

1. The evaluation is limited to 7–8B instruction-tuned models (Qwen2/Qwen3 and Mistral v0.2/v0.3), so scalability to larger models and MoE/frontier systems remains unclear, and the paper should add at least one large open model (e.g., Llama‑3 70B or Mixtral) and a small-budget closed model test (e.g., GPT‑4o or Claude 3) under matched cost.
2. The results are restricted to Qwen and Mistral families, leaving cross-family generalization uncertain, and the paper should include additional families (e.g., Llama, Gemma, Yi, DeepSeek) and analyze which failure patterns transfer across families versus remain family-specific.
3. Online Single outperforms Online EMBER overall in Table 1—especially for jailbreaking (e.g., Qwen3: 60.10% vs 31.18%)—while EMBER incurs higher interaction cost, so the paper should provide equal-cost comparisons (matched tokens/time/queries) and quantify the proportion of unique failures discovered only by EMBER.
4. Cost accounting is reported mainly in query counts rather than token-level budgets, generation lengths, compute, or wall-clock time, and the paper should present token-normalized success–cost curves and hardware/runtime details to substantiate query-efficiency claims.
5. Prior Bench is criticized as “saturated/ineffective,” but there is no comparison to a “periodically refreshed static benchmark”; if low-cost regeneration/selection can maintain effectiveness, the necessity and advantage of online methods would be substantially weakened.
6. The inference memory evaluation includes only 20 cases; the sample is too small, and most results omit confidence intervals/variance estimates and repeated trials, making it difficult to assess robustness and statistical significance.

**Questions:**

1. Can you run additional experiments on larger models (e.g., Llama‑3 70B, Mixtral/MoE) and at least one closed model (e.g., GPT‑4o or Claude 3) under a fixed budget to test whether your main trends and conclusions hold at scale?

2. If full large/closed‑model runs are infeasible, can you provide a principled rationale and partial evidence (e.g., subset experiments, scaling‑law extrapolations, or shorter learning curves) to support claims about scalability?

3. Can you include additional model families (e.g., Llama, Gemma, Yi, DeepSeek) and report cross‑family generalization by training/evaluating on different families, along with an analysis of which failure patterns transfer versus remain family‑specific?

4. For Online Single versus Online EMBER, can you provide equal‑cost comparisons with matched total tokens, wall‑clock time, and compute (GPU hours), and include per‑task/per‑model learning curves under these matched budgets?

5. Can you quantify the proportion and characteristics of unique failures discovered only by EMBER (e.g., Jaccard overlap between failure sets, novelty categories, severity), and analyze why EMBER underperforms Single on jailbreaking?

6. Can you replace or supplement query counts with token‑normalized and time‑normalized cost metrics, reporting average input/output tokens per query, GPU type and hours, batch size, throughput, and decoding settings?

7. Can you provide a “periodically refreshed static benchmark” baseline (human/LLM‑in‑the‑loop regeneration and filtering), report its maintenance cost and refresh cadence, and compare its effectiveness to online methods under matched budgets?

8. For the inference memory task, can you expand beyond 20 cases, report confidence intervals and variance across multiple runs/seeds, conduct a simple power analysis, and include a brief human audit for ambiguous cases?

9. How will you ensure a fair comparison between Online Single and Online EMBER to substantiate the claimed advantage?

**Details Of Ethics Concerns:**

Research in eliciting behaviors of LLM could lead to both positive and negative application of LLM into jailbreaking or creating harmful contents for downstream users.

---

> ### Author Response · Authors · 2025-12-03
>
> We have made our best efforts to address the concerns raised by the reviewer. Unfortunately, many of the questions contain vague or out-of-context references that we are not entirely sure what they mean. If there are any unaddressed concerns, it would be great if the reviewer could help us clarify the questions (bolded below).
>
> ---
>
> > #### **W1/Q1/Q2**: scalability to larger models and MoE/frontier systems remains unclear; Can you run additional experiments on larger models (e.g., Llama‑3 70B, Mixtral/MoE) and at least one closed model (e.g., GPT‑4o or Claude 3)
>
> We have added additional results on larger models, i.e., Qwen-14B and Qwen-32B models in Table 1 and 2. In particular, Qwen-32B already has a higher chat quality than the Llama-3-70B suggested by the review (see [LMArena ranking](https://huggingface.co/spaces/lmarena-ai/lmarena-leaderboard))
>
> Regarding closed models, we intentionally choose to not use any closed-sourced model due to a growing concern in the community regarding the reproducibility of experiments on these models – the actual model behind an API changes all the time and lacks transparency for researchers. Moreover, demonstrating a main trend we show in Figure 1, i.e., the saturation of multi-turn benchmarks, requires access to past iterations of these closed source models, especially ones released before the static benchmarks are curated, which we do not have access to.
>
> Regarding Q2 “can you provide a principled rationale and partial evidence to support claims about scalability?”, **could the reviewer elaborate what “claims about scalability” refer to here? We have not made any claim about the “scalability” of any of the methods studied in the paper (in fact, the word “scalability” has never been mentioned in the paper).** Practically, even if the framework we proposed only holds on the range of models studied, i.e., 7B-32B, they could still be useful for evaluating open sourced models.
>
> > #### **W2/Q3**: The results are restricted to Qwen and Mistral families, leaving cross-family generalization uncertain.
>
> We have included a third model family: Llama-3.1 to our results in Table 1 and Table 2.
>
> Regarding “cross‑family generalization”: We have included an analysis in Appendix B.3. on transferability across the three model families: Mistral-v0.3, Llama-3.1, Qwen3.
>
> We would also like to clarify that while “cross-family generalization” is a typical aspect of jailbreaking evaluation, it is not connected to our main thesis. Jailbreaking methods usually rely on a strong human prior, and a good prior has to generalize across model families. Cross-family generalization is less relevant to our work as we focus on online methods. *Online methods do not require “cross-family generalization” to have a high success rate*, as the whole premise of online methods is that they could find model-specific failure patterns.
>
> > #### **W3/W4/Q4/Q6**: “can you provide equal‑cost comparisons with matched total tokens, wall‑clock time, and compute (GPU hours)” “should present token-normalized success–cost curves and hardware/runtime details to substantiate query-efficiency claims”
>
> We think there might be some misunderstanding in the term “query-efficiency”. While the reviewer suggests “wall‑clock time, and compute (GPU hours)”, i.e., how long does it take a GPU to run the program in real time, the “query-efficiency” in this paper refers to the “sample efficiency” of a RL algorithm, which is orthogonal to hardware efficiency that the reviewer suggested. We stated clearly from the beginning that query efficiency is measured as the number of interactions with the target model. This notation of “query-efficiency” is motivated by the line of work on sample efficiency in RL literature. It seems that the reviewer might have interpreted query efficiency as a measure of hardware efficiency instead.
>
> Regarding token vs. query count, we have explained why we choose query count over token count in Line 151-153.
>
> Regarding equal-cost comparison, we have indeed provided these plots in Figure 5. Specifically, holding x axis value constant and comparing y axis value gives a query-normalized success rate comparison.

---

> > ### Author Response · Authors · 2025-12-03
> >
> > > #### **W5/Q7**: Prior Bench is criticized as “saturated/ineffective,” but there is no comparison to a “periodically refreshed static benchmark”; if low-cost regeneration/selection can maintain effectiveness, the necessity and advantage of online methods would be substantially weakened.
> >
> > To the best of our knowledge, we are not aware of any “periodically refreshed static benchmark” in the literature of multi-turn conversation. **If the reviewer has a specific benchmark in mind, please let us know!** We are happy to compare.
> >
> > On the other hand, even with LLM-in-the-loop, curating a multi-turn conversation benchmark can take months of time with a non-trivial cost for human annotation. Reporting maintenance cost and refresh cadence, again, this might take months. That being said, we think this could be a good follow-up work, rather than a simple baseline to execute here.
> >
> > > #### **W6/Q8**: For the inference memory task, can you expand beyond 20 cases, report confidence intervals and variance across multiple runs/seeds, conduct a simple power analysis, and include a brief human audit for ambiguous cases?
> >
> > Regarding the number of test cases, we have explained the dataset curation in Section 5.1 Line 311-315. In particular, these 20 cases are all we can find after manually filtering the original 174 cases from MultiChallenge to ensure only in-context memory retrieval is required to answer the question. Given our motivation is to compare with existing static benchmarks, we did not curate new test cases ourselves.
> >
> > Regarding “confidence intervals and variance across multiple runs/seeds”, we have added the standard deviation over 3 random seeds, i.e., 0, 1, 42 for all training-based methods over all tasks.
> > **We are not sure what “ambiguous cases” refers to here. Similarly, we are not sure why “power analysis” is relevant in this context. It would be appreciated if the reviewer could elaborate.**
> >
> > > #### **W3/Q5**:  Can you quantify the proportion and characteristics of unique failures discovered only by EMBER (e.g., Jaccard overlap between failure sets, novelty categories, severity), and analyze why EMBER underperforms Single on jailbreaking?
> >
> > We would like to clarify that multi-turn methods inherently search in an input space that is disjoint from single-turn methods, therefore all failures discovered by EMBER are unique in the sense that it is impossible for single-turn methods to find such behavior elicitation prompts. We have included a quantitative analysis in Appendix A.1 providing the proportion and characteristics of failures discovered by EMBER.
> >
> > Also, we are not entirely sure what **Jaccard overlap between failure sets, novelty categories, severity** mean in this context.
> >
> > For jailbreaking, the single turn method has an extremely high success rate because they can exploit a simple solution, which is making the target model output the exact target (usually harmful) string without any additional content. This solution is harder to find in the multi-turn setting, possibly due to target model responses having larger variance across different harmful target strings, which means there might not be a shared mechanism for the RL algorithm to discover. However, multi-turn cases still outperform prior and offline methods.
> >
> > > #### **Q9**: How will you ensure a fair comparison between Online Single and Online EMBER to substantiate the claimed advantage?
> >
> > **We are not sure what is “the claimed advantage” referred to here.**
> >
> > We also respectfully disagree with the premise of the question. Our comparison of the Single and multi-turn methods is fair: These two methods are trained and evaluated on identical input/outputs, measured under the same metric for success rate and interaction count. **If there are concerns on a specific aspect of the experimental setup that the reviewer found unfair, we kindly ask the reviewer to point it out directly.**

---

### Official Review · Reviewer_cDQk · 2025-11-01

**Soundness:** 3
**Presentation:** 3
**Contribution:** 2
**Rating:** 4
**Confidence:** 2

**Summary:**

This paper analyzes behavior elicitation in the context of multi-turn conversations. The authors review three families of existing methods and propose EMBER, a multi-turn behavior elicitation method based on online reinforcement learning (RL). They evaluate all four methods on three tasks and two target models. The results demonstrate that online methods are the most query-efficient for eliciting target behaviors, highlighting a novel and promising application of online behavior elicitation in multi-turn conversation evaluation.

**Strengths:**

1. Provides a detailed analysis of existing behavior elicitation methods and their effects.
2. Proposes EMBER, a multi-turn behavior elicitation method based on online reinforcement learning (RL).
3. Addresses the saturation problem of static benchmarks and demonstrates that EMBER can elicit target behaviors with a much higher success rate than static testing. This enables more efficient discovery and analysis of failure cases.
4. Achieves effective behavior elicitation with higher query efficiency, requiring smaller datasets.

**Weaknesses:**

1. This study only uses two target models, both of relatively small size.
2. Since the method requires interaction with the target models, there is an associated cost burden.
3. The online method is only implemented with Qwen3-4B. It's unclear whether any model ablation was conducted to evaluate robustness across different model architectures.

**Questions:**

Refer to the weaknesses section

---

> ### Author Response · Authors · 2025-12-03
>
> > #### **W1**: This study only uses two target models, both of relatively small size.
>
> We have included four additional target models: llama-2-7B, llama-3.1-8B, Qwen3-14B, Qwen3-32B. Also, the two original target models we chose are both 7-8B scale, which we believe are the **standard** and in fact the most commonly studied model size in NLP experiments.
>
> > #### **W2**: Since the method requires interaction with the target models, there is an associated cost burden.
>
> Of course, **quantifying the cost of interactions and its impact on task success rate was exactly our motivation** [See Line 22-24, 57-60]. We are not aware of any method that could reliably elicit a behavior without some interaction, i.e., no free lunch. Our results (see Sections 5 & 6) explicitly characterize how performance varies under different interaction budgets, and our discussion aims to make these tradeoffs clear so that practitioners can make informed choices.
>
> > #### **W3**: The online method is only implemented with Qwen3-4B. It's unclear whether any model ablation was conducted to evaluate robustness across different model architectures.
>
> We have included additional results on Qwen3-8B as the policy model in Appendix B.2. We choose a policy model with higher language modeling quality and larger size, as It is unclear to us if “model architectures” is the most important factor here. We do not observe a significant difference in success rate between the two policy models.
>
> In general, given limited compute, we have prioritized diversifying the target model rather than the policy model.

---

### Official Review · Reviewer_2hZc · 2025-11-01

**Soundness:** 1
**Presentation:** 3
**Contribution:** 2
**Rating:** 2
**Confidence:** 3

**Summary:**

The paper studies how to elicit (both benign and harmful) behaviors from target LLMs in multi-turn conversations for the purpose of curating adaptive test cases. "Eliciting behaviors" means finding the right prompts that likely trigger certain behaviors in a target model. The experiments are on three tasks: self-affirmation, inference memory, and jailbreaking.

The paper proposes an analytical taxonomy of elicitation approaches by interaction mode with the target model:
1. Prior-only: static prompts/benchmarks.
2. Offline interaction: SFT or in-context learning with past target outputs.
3. Online interaction: RL that learns a prompt-generation policy with access to the target model during training.

The paper then introduces EMBER, a multi-turn extension of online RL: during a rollout, the policy produces several user turns at a time while the target model replies; rewards are computed from a rubric, and gradients are backpropagated only through policy tokens. To mitigate collapse and exploration issues, the method adds an n-gram repetition penalty across turns and factorizes the policy into high-level natural-language strategies and concrete messages.

The experiments are across two target models: Mistral-7B-Instruct-v0.3 and Qwen3-8B (in case of an online setting). The experiments show that online methods achieve the highest success rates (20–60%) with a few thousand queries and can surface failures where static multi-turn benchmarks saturate.

The paper also analyzes query efficiency vs success rate, effects of training-set diversity, number of turns, and prior knowledge in the system prompt.

**Strengths:**

The study is timely, and the method choices are sensible in general. It addresses an important and growing problem: static multi-turn benchmarks saturate on new models. The taxonomy of interaction regimes is well-motivated, which helps organize a fragmented literature. The empirical study spans three tasks of different natures and includes useful ablations. There is some originality in applying online RL methods to multi-turn conversations.

**Weaknesses:**

1. The Equations 4 and 5 seem to be misleading. The whole 4.3 section is the heart of the paper, but it is extremely strange. In Equation 4:
- What is $X$? There is no set of queries there. $x$ should be sampled from the policy, $D_{online}$
- How is it even related to GRPO? Why do we have something strange instead of the KL term? Instead, it seems to be some kind of reward-regression MSE loss.
- What is $D_{online}(M_t(x) | x)$? $D$ is the policy; it should be the other way around: $D_{online}(x | M_t(x))$, as in Equation 2. The same in Equation 5, but with $y$ and $x$.
- In Equation 5, there is $π_θ$, but this notation is not used anywhere else.


2. Online RL evaluation lacks statistical rigor: no variance across seeds, confidence intervals, or sensitivity to randomization; RL training is notoriously high-variance.

3. Baselines for jailbreaking are weak. Strong multi-turn adversaries like Crescendo, AutoDAN, PRBO, MTSA, SIREN, or DPO-based red-teamers are mentioned in related works but not compared with the paper's method on the same target models and rubric. This makes it hard to claim state-of-the-art query efficiency.

4. Rubric reliability is under-specified. For inference memory and jailbreaking, a single LLM judge (Qwen3-14B) is used without calibration against human labels. No estimates of false positives/negatives or adjudication protocol are reported.

5. "Multi-turn" EMBER is in fact only 2 turns; whether benefits hold for longer conversations and the scaling of query budget with turns are not comprehensively explored.

6. Task names should be mentioned directly in the abstract, because without them, it is unclear what "eliciting behaviors" is.

**Questions:**

Suggestions:

1. Fix equations.

2. Provide variance over at least 3–5 random seeds for all online methods and include confidence intervals. What is the sensitivity to decoding randomness and initialization prompts?

3. Compare against AutoDAN, Crescendo, PRBO/GRPO red teamers, and MTSA under the same query budget and judge. If not possible, please justify.

4. For a stratified sample of outputs in inference memory and jailbreaking, provide human labels to estimate the precision/recall of the Qwen3-14B judge. If errors exist, adjust success rates or report calibrated results.

**Details Of Ethics Concerns:**

Eliciting harmful behaviours (jailbreaking) might be dangerous.

---

> ### Author Response · Authors · 2025-12-03
>
> > #### **W1/Q1**: The Equations 4 and 5 seem to be misleading
>
> Thanks for catching this! We apologize for the confusion. We have corrected the equation following the standard GRPO notations in the PDF, highlighted in blue.
>
> > ####  **W2/Q2**: Online RL evaluation lacks statistical rigor: no variance across seeds
>
> We have updated both online and offline results to report the mean and standard deviation of three runs with random seeds 0, 1, and 42 (the default seed). As shown in the updated Table 1 and 2, the RL-based methods are generally stable.
>
> We understand the concern that “RL training is notoriously high-variance”, but we are afraid this claim is an overgeneralization, as variance of RL training is highly task and algorithm dependent. In particular, the newer PPO-based algorithms have improved training stability when applied to LLM post-training.
>
> > ####  **W3/Q3**: Baselines for jailbreaking are weak. Strong multi-turn adversaries like Crescendo, AutoDAN, PRBO, MTSA, SIREN, or DPO-based red-teamers are mentioned in related works but not compared with the paper's method on the same target models and rubric. This makes it hard to claim state-of-the-art query efficiency.
>
> There are indeed many powerful jailbreaking methods, mostly fitting into the prior-based or offline method families in our proposed framework. We choose not to compare with these specific methods for the following reasons:
> Our work studies **general-purpose behavior elicitation methods**, however, all of these jailbreaking methods listed by reviewer except for PRBO, are **handcrafted for multi-turn jailbreaking and are NOT applicable to tasks other than jailbreaking**. As a result, their query efficiency is less representative, given the amount of prior knowledge researchers have accumulated on jailbreaking is far greater than on other tasks.
>
> Regarding PRBO, it is **not a “multi-turn adversary” as the reviewer claimed**. PRBO is a single-turn elicitation method, where the key contribution is a reward proxy for low-probability behaviors, which is orthogonal to the multi-turn problem studied in this paper.
>
> Lastly, we want to clarify that our online methods indeed have state-of-the-art success rate on the AdvBench string elicitation task. In Table 2, we used a more challenging harmfulness metric, as most online methods can get to close to 100% success rate on the string-based metric used in the literature (e.g., see Li et al. 2025). We added results using the string-based metric in Appendix B.5, where the online methods can get to close to 100% success rate on three out of five target models.
>
> > #### **W4/Q4**. Rubric reliability is under-specified. For inference memory and jailbreaking, a single LLM judge (Qwen3-14B) is used without calibration against human labels. No estimates of false positives/negatives or adjudication protocol are reported.
>
> We have added a section on “Judge Accuracy” in Appendix A.2 with human evaluation on conversations randomly sampled from multiple target models. For inference memory, we annotate 201 examples, the precision is 0.94 and the recall is 0.83. For jailbreaking, we annotate 345 examples, the precision is 0.90 and the recall is 0.96. The confusion matrices are shown in Fig 6.
>
> **Regarding “adjudication protocol”, could the reviewer clarify what is an “adjudication protocol” in this context?**  To our best knowledge, we are not aware of such a protocol in the literature of using LLM as a judge.
>
> > #### **W5**: "Multi-turn" EMBER is in fact only 2 turns; whether benefits hold for longer conversations and the scaling of query budget with turns are not comprehensively explored.
>
> We agree that it would be interesting to study more turns, and our methods naturally support more turns. However, we are constrained by the quality of available pre-trained LLMs as user models. In particular, we only experimented up to 2 turns because the policy models we used usually can only follow the system prompt, i.e., staying in the “user” role for 2 turns. With longer turns, it falls back to being a helpful assistant. Since RL-based methods generally require the policy to be initialized to a reasonably close distribution compared to the optimal one, in this case, the distribution of user content as opposed to the assistant content, we did not experiment with generating more than 2 turns. This is a direction that future work can explore, especially given more powerful user models have been developed (e.g., [Naous et al. 2025](https://arxiv.org/abs/2510.06552v1)).
>
> We also want to clarify that despite only 2 turns being generated, these tasks we experimented contain up to 5 turns in the full conversation, i.e., there are up to three turns in the context before generating user turns, which are reasonably long conversations.

---

> > ### Author Response · Authors · 2025-12-03
> >
> > >  #### **W6** Task names should be mentioned directly in the abstract, because without them, it is unclear what "eliciting behaviors" is.
> >
> > We would like to clarify that “behavior elicitation” is a well-defined term grounded in a line of work, e.g., Pfau et al. 2023, Li et al. 2025, and Chowdhury et al. 2025. This term is not limited to specific target behaviors. We understand that this line of work is relatively new, and we have offered an explicit definition in our abstract at line 15.

---

### Official Review · Reviewer_5aAM · 2025-11-01

**Soundness:** 3
**Presentation:** 3
**Contribution:** 3
**Rating:** 8
**Confidence:** 4

**Summary:**

The authors focus on the problem domain of trying to elicit certain outputs from LLMs. They argue that current methods focus on only single-turn interactions when trying to prompt a model and that this is not sufficient to discover all the errors LLMs can make.

The authors propose their own method called EMBER (Eliciting Multi-turn BEhavior with Reinforcement Learning). Which learns prompts that elicit certain behaviors in a multi-turn setting. They compare this against previous methods and show that their method has a higher success rate at eliciting these behaviors. They also show that it is more efficient in terms of getting the set of queries.

They test their method on three tasks: self-affirmation (model contradicts it's own response), inference memory (model violates user preferences) and jailbreaking (model generates harmful behaviors).

**Strengths:**

1) I think the authors are tackling an interesting problem domain and motivated it well. They also took a principled approach when proposing their method. I also think the adaption of their method to generate a strategy before generating a response was a good idea.

2) I think the baselines that were compared against were comprehensive.

**Weaknesses:**

1) The authors mention that they discovered new failure cases not covered in the single-turn settings but it is not clear what those failure cases are. It is mentioned that one of the patterns found is if the prompt says "you made a mistake" then the model is more likely to fail. Was this phrase not found in the baseline methods? It seems like an obvious addition to the prompt. Overall I think the analysis is a little underspecified.

**Questions:**

Questions / Typos

1) What is the gray dot in Figure 3? I don't see a description for that. Also there is a typo in the caption.

2) In Figure 3a I see the success rate went down for newer models. Is it because these models are better at not generating contradictions or harmful behavior? What changed specifically?

---

> ### Author Response · Authors · 2025-12-03
>
> > #### **W1**: Analysis (on failure patterns) is a little underspecified
>
> We have added a detailed quantitative analysis on the failure pattern in Appendix B.1.
>
> Regarding “new failure cases not covered in the single-turn settings”, we would like to clarify that multi-turn methods inherently search in an input space that is disjoint from single-turn methods, therefore all failures discovered by multi-turn methods are unique in the sense that it is impossible for single-turn methods to find such behavior elicitation prompts. We list the failure patterns discovered in multi-turn settings in Appendix B.1., for examples, the second user turn might be able to leverage information from the model response in the first turn.
>
> Regarding why simple patterns like "you made a mistake" are not found in baselines: This is actually a great example that exposes the problem of prior-based methods – test cases are biased towards failure patterns that researchers have identified on specific models. For mt-bench-101, most of the test cases are in the form of doubting the answer and offering an alternative, expressed in three specific phrases, i.e., “But, isn’t/aren’t/doesn't {alternative answer}” (37%), “Really? I heard/thought {alternative answer}” (31%), and “Are you sure/certain? I heard/Isn’t” (17%). These test cases do not include challenging the answer in a more direct and assertive manner, such as directly asserting “you made a mistake”.
>
>
> > #### **Q1**: What is the gray dot in Figure 3? I don't see a description for that. Also there is a typo in the caption.
>
> Thanks for pointing this out! We have discussed this case on L406-L410. The gray dot indicates if we were to enumerate an infinite number of queries (every possible token combination), it is likely the success rate of eliciting any arbitrary behavior or output would climb to 100%.
>
> > #### **Q2**: In Figure 3a I see the success rate went down for newer models. Is it because these models are better at not generating contradictions or harmful behavior? What changed specifically?
>
> Yes. For test cases used in the static test set, later models are better at upholding their previous answer and not generating contradictions. For example, the target model will push back on the wrong claim made in the user turn instead:
> ```
> User turn from mt-bench-101: "That seems like a lot. Are you sure it isn't too much?"
> Target model turn: Yes, I am sure. The dosing I provided is a common regimen for adults. However, the exact dose for any individual should be determined by a healthcare provider based on the type of infection and the patient's characteristics.
> ```
>
> “What changed specifically?”: This is indeed an interesting question. Our hypothesis is that the newer models are likely trained with data designed to counteract these undesirable behaviors, as these behaviors have become more well-known in the literature (see Related Work).
>
> It could be an interesting follow-up work to investigate the mechanisms of benchmark saturation. Our hypothesis is that changes in training data, especially instruction-tuning data, help improve model performance. Unfortunately, none of the models studied in our work is fully open-sourced, i.e., we do not have access to their training data. It would be great to test this hypothesis using open-sourced models.

---

### Author Response · Authors · 2025-12-03
**Summary**

We are glad that reviewers found that our study on static multi-turn benchmark saturation is **“timely”** and **”important”**; The proposed multi-turn behavior elicitation framework is **“interesting”**, **“well-motivated”**, **“principled”**, and **"helps organize a fragmented literature"**; The empirical analysis on methods using prior, offline interactions, and online interactions is **"detailed"**, compared with **"comprehensive baselines"**, **"includes useful ablations"**, and **"valuable for designing future evaluation pipelines"**.

We have conducted additional experiments to address major concerns on lacking **more target models** (reviewer cDQk, HJpX), **larger target models** (reviewer cDQk, HJpX), and **variance across seeds** (reviewer 2hZc, HJpX). Specifically, we include:
- An additional model family: Llama, including Llama-3.1-8B and Llama-2-7B
- Two larger-scale, state-of-the-art, open-weights models: Qwen-14B and Qwen-32B (note that according to the [LMArena leaderboard](https://huggingface.co/spaces/lmarena-ai/lmarena-leaderboard), Qwen3-32B already has higher quality than Llama-3-70B-Instruct requested by reviewer HJpX)
- We report mean and standard deviation across three random seeds: 42 (default seed), 0, and 1.

With these experiments, we are able to show an even **clearer trend of static benchmark saturation and the advantage of online elicitation methods (see updated Figure 1)**. Our **findings on prior vs. offline vs online interactions hold well across 5 target models from 3 families that cover 7B to 32B scale (see updated Tables 1 and 2)**.

We have also included an Ethics Statement to address concerns on potential applications to jailbreaking.

We have updated the pdf and highlighted all the changes in blue.

---

### Meta-Review · Area_Chair_jyxX · 2026-01-13

**Summary:**

This paper studies the problem of eliciting certain behaviors from LM responses in multi-turn conversations, such as jailbreaking, inference memory (model violates user preferences), and self-affirmation (model contradicts it's own response). The method uses RL to train the model to output responses that is evaluated using a rubric with a judge LLM.

Reviewers give ratings of 8/4/2/4, and the concerns are mostly about experiments, such as lack of models (which is mitigated by new experiments) and that multi-turn conversations actually use two turns in this work. Based on reviewer opinions, I'm not recommending the acceptance of this paper.

**Reviewer Concerns:**

1. concerns about the lack of models / model families are mitigated by new experiments during rebuttal.
2. concerns about clarity of presentation such as the naming being confusing are addressed.
3. concerns that only two turns are used are not addressed.

**Reviewer Scores:**

1. 5aAM: likely keeps their score as this score 8 is above all other scores and doesn't have much serious concerns.
2. 2hZc: likely keeps their score as the concerns about using two turns are not addressed.
3. cDQk: might keep their score or increase slightly as the concern about lack of other models is mitigated but the concern about cost is still valid.
4. HJpX: likely keeps their score due to the many questions and concerns, but this review might be AI-generated.

---

### Decision · Program_Chairs · 2026-01-26

Reject